# Placental network differences among obstetric syndromes identified with an integrated multiomics approach

Samantha N. Piekos [1,2,13] ✉, Oren Barak[3,4,5,13], Andrew Baumgartner[1,12], Tianjiao Chu[3,4], W. Tony Parks [6], Jennifer Hadlock [1,7], Leroy Hood[1,8,9], Nathan D. Price [1,8,10] & Yoel Sadovsky [3,4,11] ✉

The placenta is essential for pregnancy, and its dysfunction can harm both mother and fetus. To better understand placental physiology and its disruption in disease, we employ a multiomics approach (transcriptomics, metabolomics, and proteomics) combined with clinical data and histopathology from 321 placentas across conditions: severe fetal growth restriction (FGR), FGR with hypertension (FGR + HDP), severe preeclampsia (PE), and spontaneous preterm delivery (PTD). Cellular deconvolution reveals FGR + HDP placentas have more extravillous trophoblasts than controls (p < 0.0001). After adjusting for fetal sex and gestational age, we build condition-specific interomics networks and detect communities (a.k.a. subnetworks). In a control community, *miR-365a-3p* is the most connected node, whereas in FGR + HDP placentas, it is hypoxia-induced *miR-210-3p*. From this community, we identify a signature containing mRNAs implicated in placental dysfunction (e.g. *FLT1*, *FSTL3*, *HTRA4*, *LEP*, and *PHYHIP*), which distinguishes FGR + HDP placentas from those with other conditions, illustrating the power of interomics in understanding obstetric syndromes.

The placenta is essential for a healthy pregnancy, and pathology in this organ can endanger both maternal and fetal health. Unfortunately, placental dysfunction remains poorly defined[1,2]. Spontaneous preterm delivery (PTD), severe preeclampsia (PE), and severe fetal growth restriction (FGR) are among the most common obstetric syndromes; however, there are limited interventions due to a lack of knowledge of underlying mechanisms[2–4]. These conditions also have a high degree of heterogeneity in clinical presentation, disease severity, and outcomes, likely driven by different pathophysiological mechanisms[5–7]. Placental function is affected by dynamic, complex, and interconnected molecular, cellular, and environmental events[2]. We therefore need a systems biology network-driven approach to model homeostatic mechanisms and study disease in the context of disrupting normal physiological placenta function.

Data-driven, integrative network-based approaches can overcome some common challenges associated with studying pregnancy-related diseases, including the complex etiology and potential co-occurrence of these syndromes[8,9]. There are two typical approaches to integrative studies: multiple data types in the same tissue or the same data types in many tissues[10]. We have had previous successes using this approach, showing consistently that systems biology insights can be derived from correlations and network analyses across data domains and point to actionable clinical intervention[11–15]. Recently, we deployed an integrated omics approach to classify placental dysfunction patterns, improving our understanding of the pathological processes driving dysfunction[5].

We used multiple data types - placental histopathology reports, clinical and demographic features, metabolomics, proteomics, and short (micro-RNA-sequencing) and bulk (mRNA-sequencing) transcriptomics—to define interomics placenta networks specific to obstetric conditions. We then investigated the network composition and structure differences between the conditions. Next, we evaluated one community's network and

[1]Institute for Systems Biology, 401 Terry Avenue North, Seattle, WA, USA. [2]Department of Biostatistics, Epidemiology, and Informatice, University of Pennsylvania, 423 Guardian Dr, Philadelphia, PA, USA. [3]Magee-Womens Research Institute, 204 Craft Avenue, Pittsburgh, PA, USA. [4]Department of Obstetrics, Gynecology and Reproductive Sciences, University of Pittsburgh, 300 Halket Street, Pittsburgh, PA, USA. [5]Department of Obstetrics and Gynecology, Kaplan Medical Center, Rehovot, Israel, affiliated with the Hebrew University and Hadassah School of Medicine, Jerusalem, Israel. [6]Department of Laboratory Medicine and Pathobiology, University of Toronto, Simcoe Hall, 1 King's College Circle, Toronto, ON, Canada. [7]Deparatment of Biomedical Informatics and Medical Education, University of Washington, 4300 15th Ave NE, Seattle, WA, USA. [8]Buck Institute for Research on Aging, 8001 Redwood Blvd, Novato, CA, USA. [9]Phenome Health, 401 Terry Ave North, Seattle, WA, USA. [10]Thorne HealthTech, 152 West 57th Street, New York, NY, USA. [11]Pediatrics, Stanford University, 453 Quarry Rd, Stanford, CA, USA. [12]Present address: Phenome Health, 401 Terry Ave North, Seattle, WA, USA. [13]These authors contributed equally: Samantha N. Piekos, Oren Barak. ✉e-mail: samantha.piekos@pennmedicine.upenn.edu; yoels@stanford.edu

analyte level changes between conditions. The most disruption was observed in severe fetal growth restriction with pregnancy-related hypertensive disorder (FGR + HDP) placentas. We found that the analytes in this network distinguished between control and FGR + HDP placentas. These network analyses allowed us to identify molecular drivers of syndrome-specific placental dysfunction.

## Methods

### Study participants

Deidentified demographic, clinical characteristics, and stored placental biopsies were obtained from the Steve N. Caritis Magee Obstetric Maternal and Infant (MOMI) Database and Biobank at Magee-Womens Research Institute and the Health Record Research Request Service at the University of Pittsburgh as previously described[5]. Samples were biobanked from 2010 to 2019 and had placental histopathological analyses conducted by a

**Table 1 | Demographics, social history, pregnancy characteristics, and delivery characteristics of male and female fetuses**

| | | **Median (IQR)** **Female fetus** (*n* = 160) | ***n* (%)** **Male fetus** (*n* = 161) |
|---|---|---|---|
| Demographics | | | |
| Maternal age, years | | 30.0 (8.0) | 30.0 (8.0) |
| Pregravid BMI, kg/m² | | 26.8 (7.5) | 24.5 (6.3) |
| Race | Asian | 9 (5.6%) | 9 (5.6%) |
| | Black | 28 (17.5%) | 36 (22.4%) |
| | Native American | 2 (1.3%) | 3 (1.9%) |
| | White | 117 (73.1%) | 112 (69.6%) |
| | Multiracial | 0 (0%) | 0 (0%) |
| | Missing | 4 (2.5%) | 1 (0.6%) |
| Ethnicity | Hispanic | 2 (1.3%) | 2 (1.2%) |
| | Non-Hispanic | 133 (83.1%) | 135 (83.9%) |
| | Missing | 25 (15.6%) | 24 (14.9%) |
| Social history | Smoking | 21 (13.1%) | 23 (14.3%) |
| | Illicit drug use | 23 (14.3%) | 23 (14.3%) |
| Pregnancy characteristics | | | |
| Condition | Control term | 56 (35.0%) | 57 (35.4%) |
| | Severe Fetal growth restricition (FGR) | 20 (12.5%) | 16 (9.9%) |
| | FGR with pregnancy-related hypertension (FGR + HDP) | 18 (11.3%) | 12 (7.5%) |
| | Severe preeclampsia (PE) | 40 (25.0%) | 31 (19.3%) |
| | Spontaneous preterm delivery (PTD) | 26 (16.3%) | 45 (28.0%) |
| Parity | Nulliparity (0) | 74 (46.3%) | 91 (56.5%) |
| | Multiparous (1–4) | 83 (51.9%) | 70 (43.5%) |
| | Grand multiparous (≥5) | 3 (1.9%) | 0 (0%) |
| Delivery characteristics | | | |
| Delivery method | C-section | 69 (43.1%) | 58 (36.0%) |
| | Vaginal | 91 (56.9%) | 103 (64.0%) |
| Labor Initiation | Spontaneous labor | 99 (61.9%) | 82 (50.9%) |
| | Induced or no labor | 61 (38.1%) | 79 (49.1%) |
| Birthweight, g | | 2665 (1189) | 2800 (1161) |
| Gestational weeks | | 37.6 (4.0) | 37.1 (4.3) |

condition-blinded pathologist. We included placentas obtained from people ages 15–45 delivering singleton live births between 23 and 42 weeks of gestation for which the fetal sex was recorded. We evaluated five study groups (a control group and four disease groups): control-term delivery (Control; *n* = 113), PTD (*n* = 71), PE (*n* = 71), FGR (*n* = 36), and FGR + HDP (*n* = 30). Of the 30 people in the FGR + HDP group, 26 (86.7%) had preeclampsia. Control-term deliveries were defined as people whose labor started ≥37 weeks of gestation with no pregnancy complications and whose birthweight was ≥10th percentile by gestational age. PTD included people whose labor started with contractions or premature rupture of membranes and delivered <37 weeks of gestation. The severe features of PE were defined according to the guidelines of the American College of Obstetricians and Gynecologists[16]. FGR was defined as birthweight <3rd percentile for gestational age. The birthweight percentile by gestational age was calculated based on the World Health Organization's weight percentile calculator[17].

Clinical and demographic features extracted for all patients are defined and reported in Table 1 and Supplementary Table 1. Differences in cohort composition based on fetal sex were evaluated using a Student's two-tailed *t*-test, chi-square test, or Fisher's exact test for continuous, categorical, and binary variables, respectively (Supplementary Table 2). Any categories containing a zero value for either cohort were dropped before completing a chi-square or Fisher's exact test.

Participants in this study (protocol #20040257) delivered at the Magee-Womens Hospital (MWH), Pittsburgh, Pennsylvania, and provided informed consent for placental collection under three complementary protocols, all approved by the institutional review board at the University of Pittsburgh (protocols #19100240, #19120076, and #19100330). All ethical regulations relevant to human research participants were followed.

### Omics data generation

Placental metabolites were analyzed by Metabolon (Morrisville, NC) using the Global Metabolomics platform as previously described[5]. Metabolon's informatics system was used for data extraction and peak identification, compound identification and quantification, curation, and data normalization. Samples were randomized across several batches and processed with pooled quality-control samples in each batch. Across all biological samples, 1032 metabolites were detected, 925 of which received chemical annotation (Supplementary Table 3).

As previously described, placental proteins were extracted and analyzed using five Olink Target 96 panels (Olink Proteomics): Cardiovascular II, Cardiovascular III, Development, Inflammation, and Oncology III[5]. We measured 452 unique proteins in each placenta (Supplementary Table 3). Samples were processed in batches with pooled quality-control samples, which were included in each batch. All assay-validation data (detection limits, intra-, and inter-assay precision data) are available on the manufacturer's website (www.olink.com).

RNA was isolated from placental biopsies, and total and small RNA-seq libraries were prepared and sequenced as previously described[5]. Total RNA-seq libraries were sequenced on an Illumina NovaSeq 6000, using an S4 200 flow cell (Illumina, #20028313), with read lengths of 2 × 101 bp and an average of ~40 million reads per sample. Small RNA-seq libraries were sequenced on an Illumina NextSeq 2000, using a P3 50 flow cell (Illumina, #20046810) with read lengths of 1 × 75 bp and an average of ~12 million reads per sample. The RNA libraries were aligned to the human reference genome GRCh38 using the RNA-seq alignment tool STAR[18] and annotated with GENCODE 30[19]. We used STAR quantMode GeneCounts to calculate the reads per gene for each RNA-seq library[20]. Per placenta, 51,174 mRNAs and 2414 miRNAs were detected by total and small RNA-seq, respectively (Supplementary Table 3).

### Omics data preprocessing

All analytes were filtered for <20% missingness with missing values imputed as half the minimum value measured for that analyte, and for >50% of unique measured values, ensuring that the analyses included only variable analytes. For total RNA-seq data, mRNA with <500 counts were excluded.

For small RNA-seq data, miRNA <10 counts were excluded. Post-processing 865 metabolites, 343 proteins, 448 miRNAs, and 9582 mRNAs were included in the subsequent analyses (Supplementary Table 3).

## Deconvolving bulk RNA-seq data

We performed deconvolution of our bulk placenta RNA sequencing with the CIBERSORTx web application[21]. A cell-type signature matrix from ref. 22 was generated using a single-cell RNA sequencing reference sample from healthy term villous tissue. This data were normalized to counts per million (CPM), batch corrected using "S-mode", and permuted 100 times to assess statistical significance. Cell-type compositions $x$ live in the n-simplex, $\Delta^n$, defined as

$$\Delta^n = \left\{ x \in R^n \,\middle|\, \sum_{\{i=1\}}^{n} x_i = 1, \, x_i > 0 \right\}.$$

The constraint $\sum_{\{i=1\}}^{n} x_i = 1$ induces an inherent geometry on the space $\Delta^n$ known as the Aitchison geometry[23]. The clr transformation, defined as

$$clr(x_i) = ln \frac{x_i}{g(x)}$$

with

$$g(x) = \prod_{\{i=1\}}^{n} x_i^{\{\frac{1}{n}\}}$$

is a map $\Delta^n \rightarrow R^n$ that preserves distances in the Aitchison geometry. Since most statistical techniques are primarily used for Euclidean data, this map allows us to perform statistical analyses that better respect the compositional nature of transcriptomics data. Campbell et al.[22] tested this deconvolution approach using bulk RNA sequencing data from both healthy term and preeclampsia placental samples, which included earlier gestational ages.

We performed the clr transformation with custom functions built using numpy v1.24.3, with zeros imputed additively with a value of $10^{-6}$, followed by renormalization of $x$. We ran Kolmogorov–Smirnov tests between the control and each condition (FGR, FGR + HDP, PE, and PTD) for each clr-transformed cell type composition using scipy.stats v1.10.0. We then performed Benjamani–Hochberg multiple hypothesis correction via statsmodel v0.13.5. Boxplots reported the difference in cell type proportions between control and obstetric syndrome placentas. The cell type heatmap with hierarchical clustering in Fig. 1A was made using seaborn v0.12.2. The row and cluster dendrograms were made using clr-transformed data with scipy.cluster.hierarchy v1.10.0.

## Adjusting for fetal sex and other common covariates

We controlled for common covariates by training generalized linear regression models (GLM) for each analyte for each fetal sex (GLM: analyte ~ weeks gestation at delivery + pregravid $BMI^2$ + C(Labor Initiation) + C(Smoking Use) + C(Illicit Drug Use)). Maternal age is anticorrelated with gestational age (Supplementary Fig. 8), and the two variables have a high level of multicollinearity. Therefore, we decided not to include maternal age alongside gestational age in our GLMs. We trained GLM models for each fetal sex (male $n = 161$; female $n = 160$) because this better accounts for the impact of fetal sex on a placental analyte behavior than training one GLM model and accounting for fetal sex by adjusting the y-intercept. We performed correlation analysis to select the variables included in these models. We confirmed that multicollinearity was minimized by the variance inflation factor (VIF) analysis.

We performed an Anderson–Darling test to evaluate the normality of each analyte's distribution. Z-score was used to detect outliers with the threshold set as the absolute value of three, if more than 2.5% of all measurements surpassed this threshold, then the analyte was declared an outlier.

We used Scipy v.1.10.1 to perform these tests. A Gaussian family was used to train the GLM unless an analyte had a non-normal distribution ($p < 0.05$) or had outliers, in which case a gamma family was applied. We trained the GLMs using statsmodels v.0.14.0 and then performed Benjamini–Hochberg multiple hypothesis correction. The GLMs were used to adjust for covariates of the analytes significantly regulated by gestational days at delivery. The gestational age at delivery beta-coefficients for the significant analytes in common between the fetal sexes was visualized in a heatmap.

Random simulation was performed to calculate the false discovery rate (FDR) for the number of analytes regulated in both fetal sexes compared to random chance. This simulation approach effectively creates a null distribution, representing what the overlap of gestational age-regulated analytes would look like if there were no true sex-specific differences. By comparing our actual results to this null distribution, we can assess the statistical significance of our findings and estimate the FDR. To do this, we performed a hundred iterations of randomly assigning the samples to one of two groups and then calculating the GLMs for each group's analytes as described above. For each iteration, we tracked the number of analytes significantly regulated by gestational age at delivery following Benjamini–Hochberg correction in each group and those in common between the two groups. The FDR was calculated as the number of iterations with fewer analytes shared between the two random groups than the number reported for both fetal sexes. Additionally, we reported the FDRs of the proportion of analytes for a random group that overlapped with the second random group. These random simulation distributions were reported as histograms.

## Correlation network and community analysis

For each pairwise datatype (e.g., metabolomics vs. placental histopathological features, metabolomics vs. proteomics, etc.) each measurement from the first dataset was correlated with every measurement from the second dataset using Spearman's $\rho$. These correlations were performed on analyte measurements adjusted for gestational age as a function of fetal sex as described in the previous Methods subsection. This was performed in all five conditions for all interomics correlations among the placental histopathological features, metabolomics, proteomics, short transcriptomics, and total transcriptomics data (16,705,671 associations per condition). Bonferroni multiple hypothesis correction was performed on a chosen significance level of $p < 0.05$. The top 50 most significant associations for each datatype for each condition were visualized in circus plots by nxviz v.0.7.3. These tests were repeated while downsampling the other conditions to the smallest group size (FGR + HDP; $n = 30$) to evaluate the impact of power on the number of associations detected. The associations for each condition calculated under full power were used in subsequent analyses. Histograms of the absolute value of the Spearman coefficients were plotted for each condition, calculated at full power and by downsampling to evaluate the effect size of significant interomic correlations.

We performed Louvain community detection (resolution = 0.45) to define subnetworks within each obstetric phenotype; a greedy optimization method that iteratively removes edges while maximizing modularity[24]. The quality of the partitions was evaluated using modularity, performance, and coverage. This was completed using NetworkX v.3.1. Closeness centrality, a measure of network connectivity calculated by the average shortest path for a node in the network to all other nodes, was calculated by NetworkX. This was visualized in heatmaps with hierarchical clustering. Communities were visualized by pyvis v.0.3.1 with blue edges indicating positive correlations and red edges indicating negative correlations. The color of the nodes corresponded to the closeness centrality. We evaluated differences in network structure by examining the community size and average shortest path, which were calculated using NetworkX v.3.1. We used the Jaccard Similarity Index to evaluate network composition. These statistics were reported in dot and whisker plots.

## Statistics and reproducibility

To evaluate enrichment for transcripts and proteins within subgroups, gene ontology (GO) biological process terms analysis was performed using

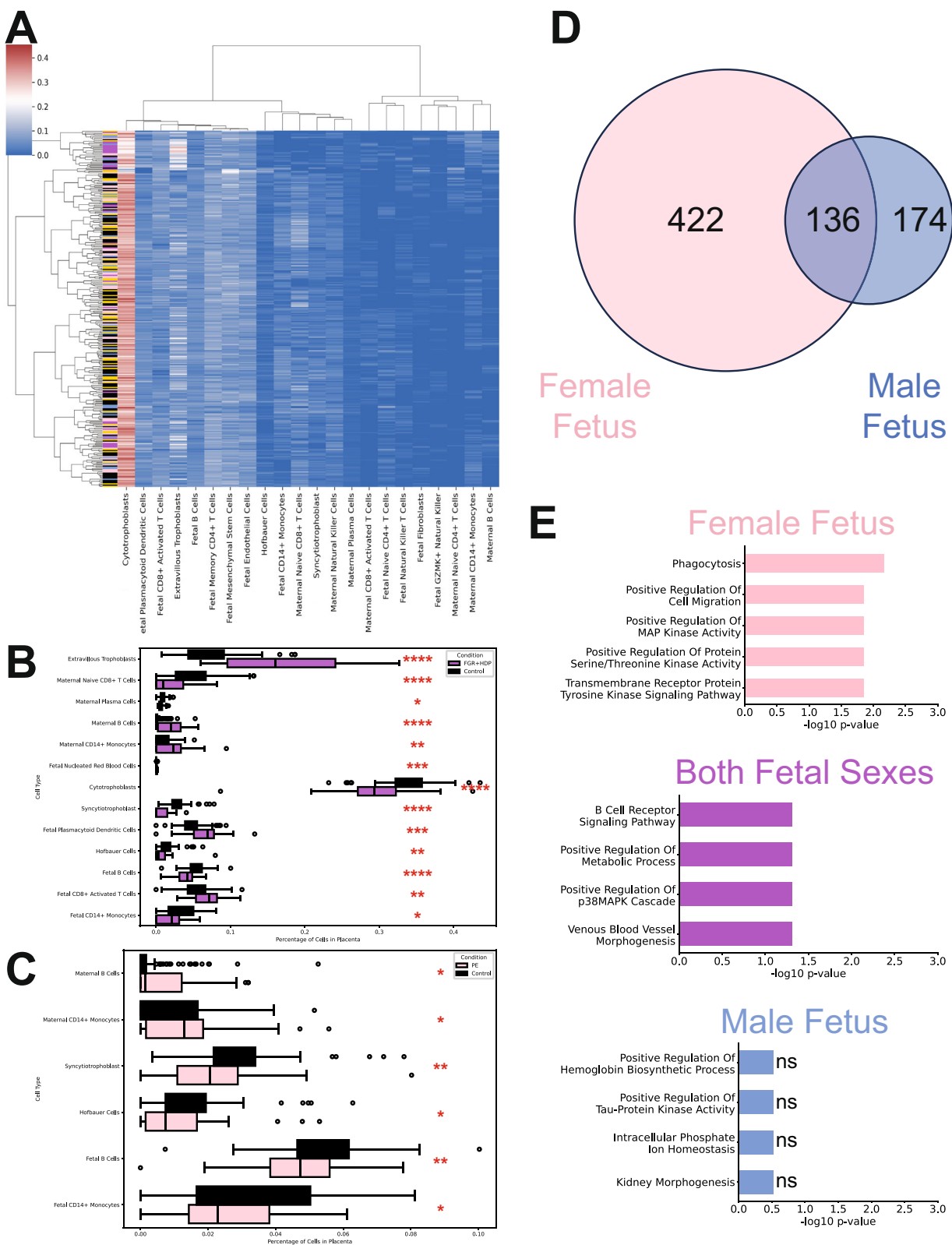

Enrichr[25]. Results were visualized in bar charts as −log10($q$ value). Differential analyte levels between pairs of conditions were calculated on log2 fold-change (FC) data using a student's two-tailed $t$-test with a Benjamini–Hochberg correction. Only analytes with significant differential expression with at least 1.5 FC for at least one pairwise comparison were visualized in a heatmap with hierarchical clustering.

Principal component analysis (PCA) and partial least squares-discriminant analysis (PLS-DA) were unsupervised and supervised learning approaches, respectively, which were used to evaluate the separation between conditions in high-dimensional space using adjusted analyte values. The adjusted analyte values were divided by the average value for the analyte before PCA or PLS-DA analysis to account for the overall expression

**Fig. 1 | Placental cellular composition and fetal sex differences were detected.**
**A** Cell type cluster map of the percentage of cell types (x-axis) composing each placenta (y-axis). The color indicates the percentage of the cell type in each placenta, with blue indicating a low percentage and red indicating a high percentage. Hierarchical clustering was performed to group both the cell types and placentas. A color bar was provided along the y-axis to indicate the condition of each placenta: control (black; $n = 113$), FGR (blue; $n = 36$), FGR + HDP (purple; $n = 30$), PE (pink; $n = 71$), and PTD (gold; $n = 71$). **B, C** Boxplots of significantly different cell type percentages in control compared to FGR + HDP (purple; **B**) and PE (pink; **C**) placentas. Each box displays the interquartile range (IQR) from the 25th percentile (Q1) to the 75th percentile (Q3). The line in each box is the median. Whiskers extend to 1.5 times the

IQR, and the circles outside the whiskers are outliers. The $p$ values were calculated by Kolmogorov–Smirnov tests followed by Benjamini–Hochberg multiple hypothesis corrections. *$p < 0.05$, **$p < 0.01$, ***$p < 0.001$, ****$p < 0.0001$. **D** Venn diagram of the number of placental analytes significantly regulated by gestational age in female fetuses (pink; $n = 160$), male fetuses (blue; $n = 161$), or both sexes (purple) as determined by GLMs followed by Benjamini–Hochberg multiple hypothesis correction. **E** Top Gene Ontology Biological Process (GO BP) Terms for mRNA transcripts and proteins significantly regulated by gestational age in female fetuses (pink, top), both sexes (purple, middle) or male fetuses (blue, bottom). The adjusted $p$ value (Benjamini–Hochberg correction) was reported in −log scale on the x-axis.

signal. Each dot plotted represents a placenta, and the dot color indicates the clinical condition. FGR + HDP placentas were compared to the placentas of each condition in pairwise comparisons.

### Data visualization
We visualized the heatmaps with and without clustering, histograms, boxplots, dot plots, PCA plots, and PLS-DA plots using Matplotlib v3.8.2. In boxplots the box represents the IQR from the 25th percentile (Q1) to the 75th percentile (Q3), the line in the box represents the median, and whiskers extending beyond the box to 1.5 times the IQR. Outliers that exist beyond the whiskers are plotted as circles. In dot and whisker plots the mean and standard deviation represent the dot and whiskers respectively.

### Reporting summary
Further information on research design is available in the Nature Portfolio Reporting Summary linked to this article.

## Results
### FGR + HDP had the greatest difference from control in placental cellular composition
We evaluated the cellular composition of the placentas by performing single-cell deconvolution using the bulk RNA-seq data (Fig. 1A and Supplementary Fig. 1). No differences in cellular composition were detected between FGR or PTD and control placentas (Supplementary Figs. 2, 3 and Supplementary Table 4, 5). There were significant differences in the proportion of a subset of cell types in both FGR + HDP and PE compared to the control (Fig. 1B, C, Supplementary Figs. 4–7, and Supplementary Tables 6, 7). FGR + HDP placentas were the most distinctive with the largest significant increase observed in extravillous trophoblast cells compared to control placentas (Fig. 1B and Supplementary Table 6). There were also significantly elevated levels of fetal CD8+ activated T cells, plasmacytoid dendritic cells, fetal nucleated red blood cells, maternal CD14+ monocytes, and maternal B cells as well as depleted levels of fetal CD14+ monocytes, fetal naive CD4 + T cells, fetal B cells, Hofbauer cells, syncytiotrophoblast, cytotrophoblasts, maternal plasma cells, and maternal naive CD8 + T cells in FGR + HDP compared to control placentas (Fig. 1B, Supplementary Fig. 4, and Supplementary Table 6).

### More placental analytes are associated with gestational age in female fetuses
We performed a correlation analysis of common confounders from which we chose five variables to include in the GLMs (Supplementary Fig. 8). The resulting VIF confirmed limited multicollinearity (Supplementary Table 8). For each fetal sex, GLMs were trained for each analyte as a function of gestational age at delivery, pregravid BMI, labor initiation status, smoking status, and illicit drug use status. About 732 analytes were significantly regulated by gestational age at delivery in one (422 female only, 136 male only) or both fetal sexes (174 both sexes; Fig. 1D and Supplementary Data 1, 2). The effect size of gestational age on placental analyte levels was small and remained consistent in both size and directionality for analytes regulated by both fetal sexes (Supplementary Figs. 9, 10). There was a significantly lower number of analytes in common between male and female fetuses than

expected by random chance (FDR <0.01; Supplementary Fig. 11A). The proportion of analytes significantly regulated by gestational age at delivery shared between female and male fetuses was lower than random chance (FDR <0.02; Supplementary Fig. 11B). However, the proportion of significant analytes in male fetuses that were in common with female fetuses was not different from that expected by random chance (FDR = 0.14; Supplementary Fig. 11B). The GLMs were used to adjust these significant analytes for the following cofactors: gestational age at delivery, pregravid BMI, labor initiation status, smoking status, and illicit drug use status.

We evaluated gene ontology (GO) terms for the proteins and mRNAs that were significantly regulated in female fetuses only, both sexes, or male fetuses only (Fig. 1E). There was some overlap in female fetus-specific and both sexes analytes, such as positive regulation of map kinase activity. However, there are also differing terms associated with significantly regulated analytes in female fetuses only and both sexes, such as phagocytosis and venous blood vessel morphogenesis, respectively. No GO terms were associated with proteins and mRNA male fetus-specific analytes.

### Community structure in interomics correlation networks
After adjusting placental analyte measurements for gestational age as a function of fetal sex, we built five interomics correlation networks from Spearman correlations across the five conditions (Fig. 2 and Table 2). In these networks, nodes correspond to analytes, and an edge exists between two nodes if there is a significant correlation ($p < 0.05$) following Bonferroni correction. The effect size was large for all significant correlations in all conditions as assessed by the Spearman coefficient (Supplementary Fig. 12). There were more significant correlations in the disease networks for PTD, PE, and FGR + HDP by two orders of magnitude compared to the control network after downsampling to $n = 30$ to allow evaluation at the same power level (Supplementary Table 9). There was a more limited number of significant correlations in the FGR network, which limits interpretation in downstream analysis for the FGR network. Finally, we found no significant correlations in any network when we focused on the placental histopathological features.

We performed Louvain community detection to define subnetworks within each obstetric phenotype by iteratively removing edges while maximizing modularity. We reported the number of communities (a.k.a. subnetworks) per obstetric type (Table 2). The modularity, performance, and coverage were reported to evaluate the degree of network structure and the quality of the partitions (Table 2). The number of communities detected was higher in Control placentas than in the other conditions (Table 2). The community sizes were also smaller in Control placentas than those detected in PE, PTD, and FGR + HDP placentas (Supplementary Fig. 13A). We evaluated overlap in the node identity between the obstetric networks using the Jaccard similarity index (Supplementary Fig. 13B). PE and PTD had the highest level of overlap with a score of 0.39 indicating that 39% of the analytes are shared between the two networks, which may suggest some commonality of underlying biological pathways. (Supplementary Fig. 13B). Control had a moderately low level of overlap with PE and PTD, with scores of 0.13 and 0.15, respectively. FGR + HDP had a low level of overlap with any other condition (Supplementary Fig. 13B).

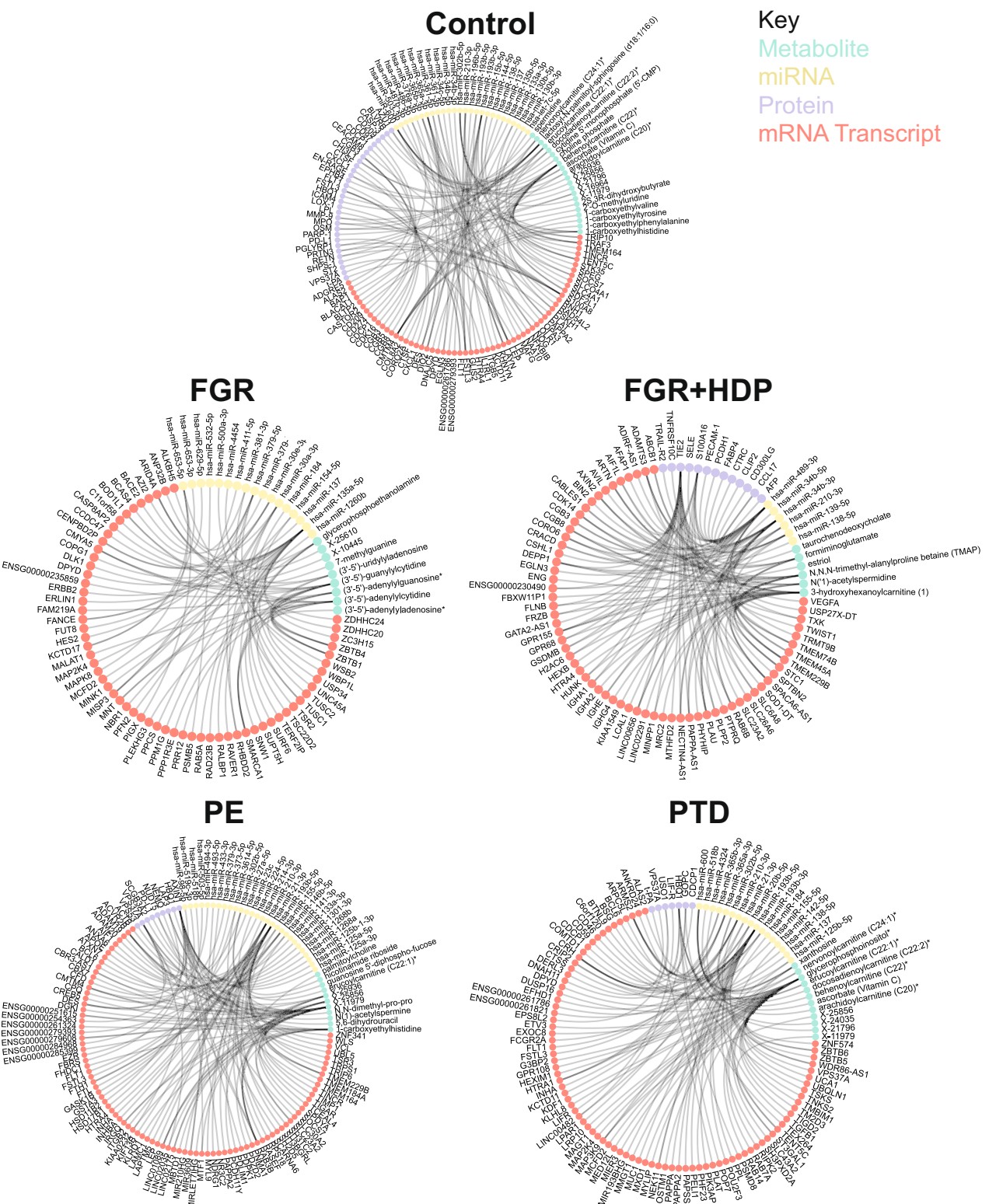

**Fig. 2 | Top 50 correlations by datatype by condition.** The subset of top statistically significant Spearman interomic cross-sectional correlations between all datasets collected in each cohort. Each line represents one correlation that was significant after adjustment for multiple hypothesis testing using the method of Bonferroni Correction. The top 50 most significant correlations for metabolite (green nodes), miRNA (gold nodes), protein (purple nodes), and mRNA transcript (orange nodes) are displayed in a circos plot. No significant correlations were found between placental histopathological features and any analyte type for any condition. The number of samples per obstetric syndrome are as follows: Control ($n = 113$), FGR ($n = 36$), FGR + HDP ($n = 30$), PE ($n = 71$), and PTD ($n = 71$). The number of features per datatype were analyzed: metabolites ($n = 865$), miRNA ($n = 448$), mRNA transcripts ($n = 9582$), placental histopathological features ($n = 13$), and proteins ($n = 343$). A total of 16,705,671 Spearman correlations were made between features for each obstetric syndrome.

**Table 2 | Network and partition quality metrics and overview**

|  | Control | FGR | FGR + HDP | PE | PTD |
|---|---|---|---|---|---|
|  | *n* = 113 | *n* = 36 | *n* = 30 | *n* = 71 | *n* = 71 |
| Edges (*n*) | 2267 | 97 | 80,612 | 88,795 | 18,443 |
| Nodes (*n*) | 1137 | 98 | 714 | 7941 | 4088 |
| Communities (*n*) | 38 | 14 | 2 | 18 | 12 |
| Modularity | 0.8 | 0.9 | 0.55 | 0.62 | 0.77 |
| Coverage | 0.95 | 0.99 | 1 | 0.84 | 0.99 |
| Performance | 0.73 | 0.88 | 0.32 | 0.42 | 0.58 |

The control placental network modularity score was 0.8, which is very high and indicates a strong community structure with effective clustering and defined community boundaries (Table 2). This was higher than in PE, PTD, and FGR + HDP placental networks. PTD had the next highest modularity score of 0.77, indicating a high level of community structure (Table 2). PE and FGR + HDP have modularity scores of 0.62 and 0.55, respectively, reflecting a moderate network community structure with fairly effective clustering but some degree of interaction between the communities (Table 2). Coverage scores indicated the percentage of nodes assigned to a community. Coverage scores were high for all conditions except pre-eclampsia (PE), indicating successful partitioning of nodes into communities and a robust community structure in these networks (Table 2). PE had a moderately high coverage score of 0.84, but with 16% of nodes unassigned, suggesting that while the community structure in PE is generally well-defined, it is worse than the other outcomes (Table 2).

The performance score indicated how well the Louvain community detection algorithm optimized the network partitioning to maximize modularity. The Control performance score was 0.73, indicating a strong community structure with well-defined communities and a relatively high level of algorithm optimization (Table 2). PTD had a performance score of 0.58, indicating a moderate optimization level with successful community identification (Table 2). However, respective performance scores of 0.42 and 0.32 in PE and FGR + HDP networks indicated suboptimal algorithm performance and lower definitions between communities (Table 2). To evaluate the interconnectivity structure of communities within the different conditions' networks, we examined the average shortest path and average clustering coefficient within communities in each network. This was similar to Control, PTD, and PE networks, but substantially lower in the FGR + HDP network, suggesting a noticeably higher level of connectivity within FGR + HDP communities than in the other conditions (Supplementary Fig. 13C). The average clustering coefficient in FGR + HDP (0.494) was 29 times higher than the next highest average clustering coefficient observed in the control network (0.017; Supplementary Fig. 13D).

**Mining multiomic communities for biological signatures**

We examined each of the communities in the Control and corresponding communities in the obstetric syndromes (Supplementary Figs. 14–18 and Supplementary Data 3–7). One community in Control placentas consists of 100 analytes, including several previously implicated in placental dysfunction, including *FLT1*, *FSTL3*, *HTRA4*, *LEP*, and *miR-210-3p* (Fig. 3A, B). All of these analytes had moderate levels of network connectivity, as measured by closeness centrality scores (Fig. 3B). However, the most connected node in this subnetwork was *miR-365a-3p* (Fig. 3B). Closeness centrality scores of these 100 analytes in the other conditions indicate a substantial disruption in network connectivity, particularly in FGR + HDP placentas (Fig. 4A). Only 16/100 analytes participated in any significant interomics correlations in FGR + HDP placentas, and the most connected node in the corresponding community was *miR-210-3p* (Fig. 4A).

Furthermore, these analytes were substantially upregulated in FGR + HDP placentas compared to all other conditions (Fig. 4B). Hierarchical clustering of differential expression across all pairs of conditions revealed that the analyte with the most distinctive expression changes was *HTRA4*

(Fig. 4B). The cluster with the second most distinctive expression profile included *DIO2*, *EGLN3*, *FLT1*, *FSTL3*, *GPRC5A*, *LEP*, *NOG*, *NOTUM*, *PHYHIP*, *SFRP1*, and *TIMP3* for a total of 12 mRNA transcripts. The 100 analytes were enriched for genes involved in hematopoietic cell proliferation (Fig. 5A). However, these 12 analytes, a subset of the cluster, were implicated in different processes, including apoptosis and Wnt signaling (Fig. 5B).

Analytes belonging to this control community, especially the 12 analytes, displayed a distinctive expression profile in FGR + HDP placentas, and we tested if we could use them to separate Control and FGR + HDP placentas in high-dimensional space. Using all metabolites, proteins, miRNAs, or mRNAs failed to distinguish between FGR + HDP placentas from other conditions by PCA (Supplementary Fig. 19). However, limiting the analytes to the 100 analytes in the Control community showed a separation between FGR + HDP and other condition placentas accounting for 63.5%-66.1% of the variance in the data across the first two principal components for each pairwise comparison (Fig. 5C). Repeating these PCA analyses with the 12 analytes with the most different expression patterns across all conditions achieved better levels of separation between FGR + HDP and other condition placentas (Fig. 5D). It also accounted for higher levels of variance in the data for the first two principal components across all comparisons (84.0%-85.0%). The separation between FGR + HDP and other placental conditions was more extreme with the supervised learning approach PLS-DA (Supplementary Fig. 20). Therefore, these 12 analytes may serve as a biomarker signature for FGR + HDP placentas at delivery time.

## Discussion

We observed that FGR + HDP placentas exhibited the most pronounced differences compared to other obstetric conditions. Notably, the cellular composition of FGR + HDP placentas differed from controls, with a significant increase in extravillous trophoblast cells. Additionally, we detected small fetal sex differences in analytes significantly regulated by gestational age, finding fewer common analytes between sexes and a higher number of female-specific analytes than expected by random chance. We identified thousands of statistically significant interomic correlations, which we partitioned into communities to organize biomarkers within condition-specific biological networks. Among these, the FGR + HDP network was the densest and least structured compared to other phenotypes. We identified twelve analytes associated with apoptosis and Wnt signaling, forming a distinct signature for FGR + HDP placentas. Overall, using an interomics network approach allowed us to define biological network differences across conditions and to apply dimensionality reduction. This allowed us to define a specific signature for FGR + HDP. These findings enhance our understanding of the placental function.

This study included many samples across various obstetric conditions with multiple data types. This enables a hypothesis-free approach to interrogating the distinction between heterogeneous conditions. Even after applying stringent multiple-hypothesis testing, we consistently detected the highest signal level in FGR + HDP placentas, making them the most distinct from all other conditions. This makes sense as this condition is the most severe of the clinical outcomes examined here. A smaller but still statistically and biologically significant signal was detected in FGR, which likely reflects the high level of heterogeneity within this condition. This finding supports that FGR with pregnancy-related hypertension should be considered distinct mechanistically from FGR without hypertension and studied as an independent condition.

In line with Campbell et al. 2023, we observed a reduced presence of Hofbauer cells in the PE placentas[22]. However, unlike the previous study, we detected a significant increase in extravillous trophoblast cells in FGR + HDP placentas, with no such increase in PE placentas[22]. This may be due to birth weight at delivery not being accounted for in the Campbell et al. 2023 inclusion criteria, which could allow PE samples from small for gestational age babies to drive this result[22]. Regardless, this difference highlights the importance of more granular clinical definitions when handling these biologically heterogeneous and clinically overlapping conditions. Furthermore, it is necessary to take fetal sex into account when considering obstetric

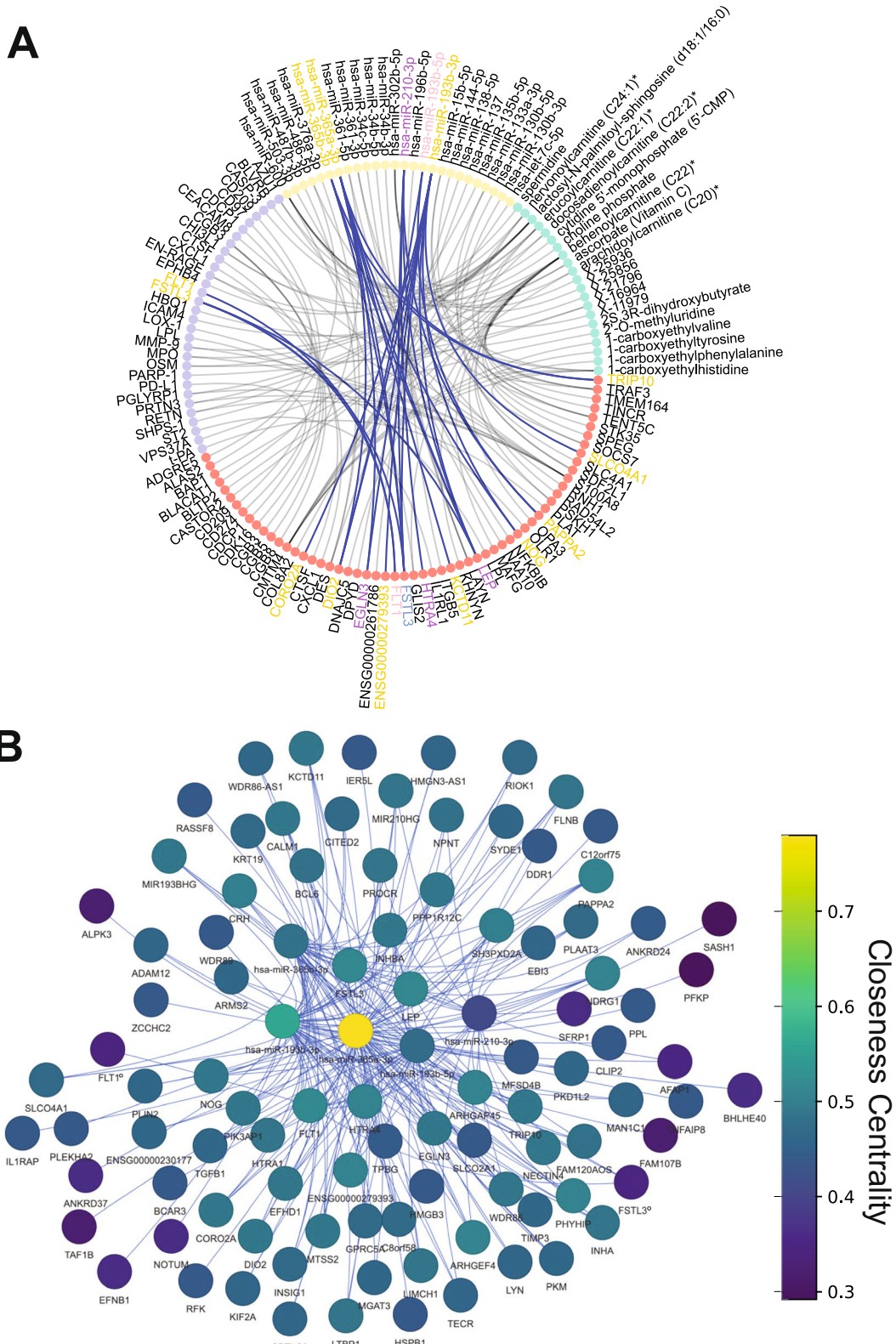

**Fig. 3 | *FLT1* and *FSTL3* community. A** Circos plot of the top 50 most significant correlations for metabolites (green nodes), miRNA (gold nodes), protein (purple nodes), and mRNA transcript (orange nodes) in the control interomics network made from 113 control placental samples. The highlighted edges indicate associations in the *FLT1 / FSTL3* community, with blue edges indicating positive correlations. The color of the text of the node name indicates whether the analyte level was adjusted for gestational age in female fetuses (pink), male fetuses (blue), both sexes (purple) or neither (gold) before interomics network analysis. **B** *FLT1 / FSTL3* community network graph. Node color indicates closeness centrality score, with gold being the most connected and purple the least connected nodes in the graph. The edge color indicates the directionality of the correlation, with blue indicating a positive correlation. Nodes are labeled (*n* = 100). ° at the end of a node name indicates that the node is a protein.

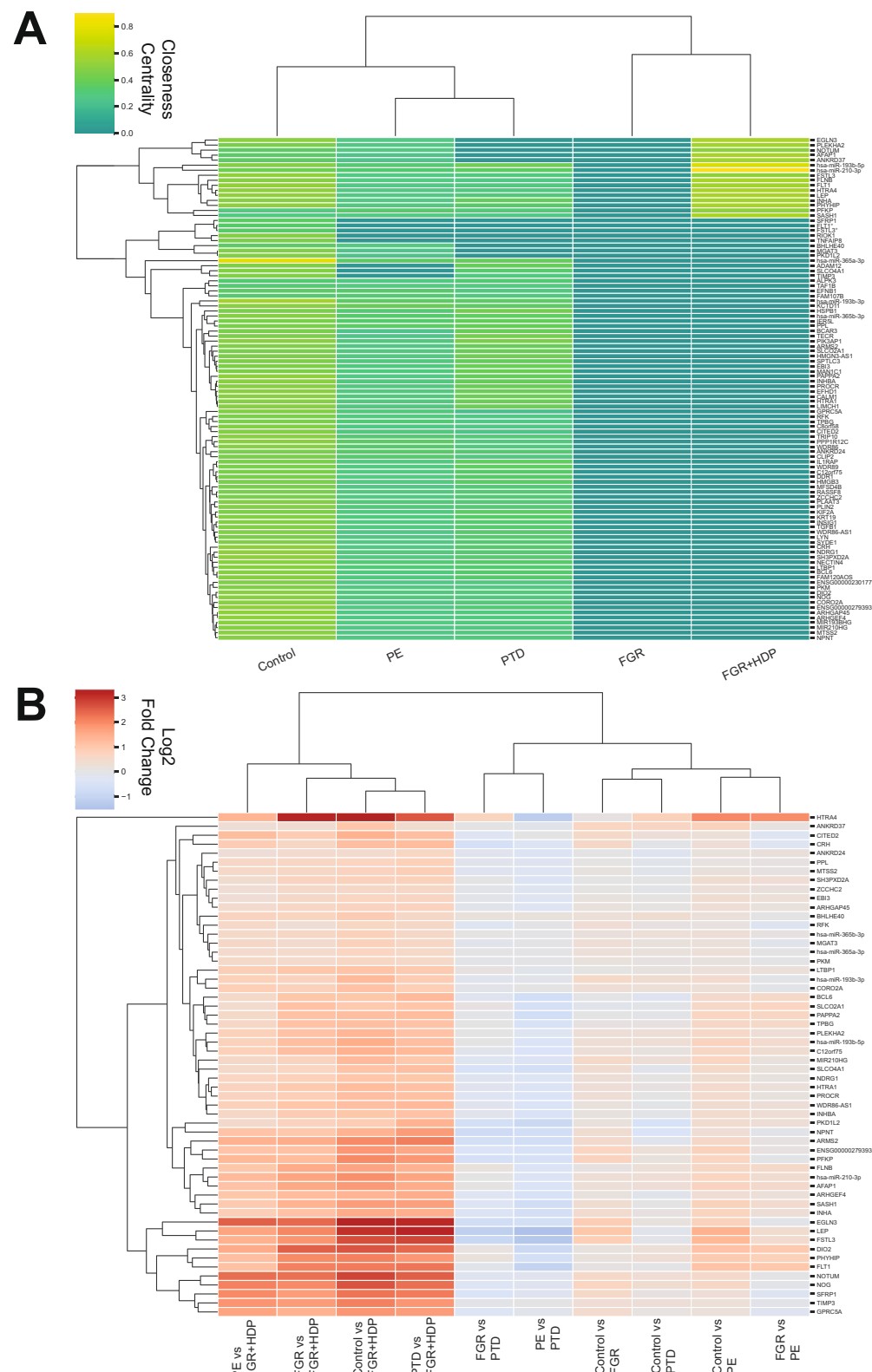

**Fig. 4 | Network connectivity and expression levels are substantially different for the *FLT1 / FSTL3* community in FGR + HDP placentas. A** Cluster map of the closeness centrality score for the 100 analytes in the control *FLT1 / FSTL3* community for all five conditions. The color indicates the level of connectivity, with gold indicating a high level and purple a low level of connectivity. Hierarchical clustering was performed to group both the conditions and the analytes. **B** Cluster map of the differential expression in pairwise comparisons between all conditions. The log2 fold-change (FC) is plotted with red indicating a high level and blue a low level of

expression change. Only analytes with at least 1.5 FC and adjusted *p* value of ($p_{adj} < 0.05$) in at least one pairwise comparison were included. FC was calculated using a Student's two-tailed *t*-test and Benjamini–Hochberg was used for multiple hypothesis correction. Hierarchical clustering was performed to group both the pairwise comparisons and the analytes. The number of samples per obstetric syndrome are as follows: Control ($n = 113$), FGR ($n = 36$), FGR + HDP ($n = 30$), PE ($n = 71$), and PTD ($n = 71$).

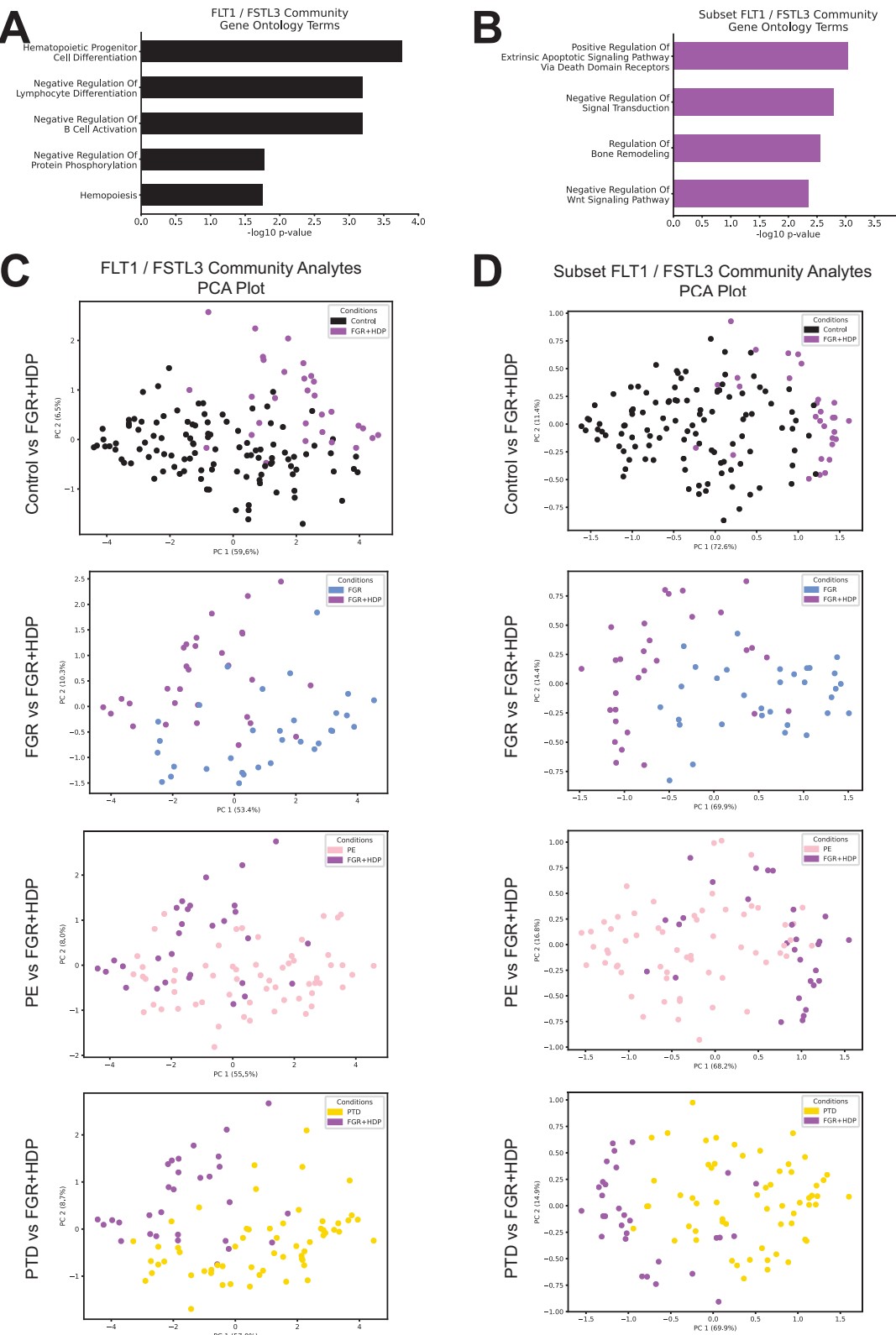

**Fig. 5 | A unique signature for FGR + HDP placentas is defined. A**, **B** Top Gene Ontology Biological Process (GO BP) Terms for mRNA transcripts and proteins for all 100 analytes in the *FLT1* / *FSTL3* community (**A**) and the twelve mRNA transcripts used in the final FGR + HDP biosignature (**B**). The adjusted *p* value (Benjamini–Hochberg correction) was reported in −log scale on the x-axis. **C**, **D** Principal component analysis of the FGR + HDP (purple; *n* = 30) placentas plotted alongside the control (black; top panels; *n* = 113), FGR (blue; middle top panels; *n* = 36), PE (pink; middle bottom panels; *n* = 71), and PTD (gold; bottom panels; *n* = 71) for both all 100 analytes in *FLT1* / *FSTL3* community (left; **C**) and the twelve mRNA transcripts in the final FGR + HDP biosignature (right; **D**). Each dot represents a placenta, with the color of the dot indicating the placenta's condition. The twelve analytes in the final FGR + HDP biosignature were selected for having the most distinctive differential expression across all pairwise comparisons of conditions as revealed by hierarchical clustering (Fig. 4B).

syndromes. Here, we observed sex-related differences in analyte levels, especially in female fetuses. This might reflect greater fetoplacental adaptability and reserve capacity in female fetuses, enabling a more robust response to the in-utero environment[26]. Sex-specific differences in innate and adaptive immunity have been linked to placental health and fetal growth[27].

We defined interomics networks for each obstetric condition. After adjusting for power, there were four orders of magnitude more significant interomic correlations for PE and PTD compared to the control and five orders of magnitude more for FGR + HDP. The networks associated with these syndromes are more densely packed, less organized, and composed of fewer, larger, and more poorly defined communities. This suggests a higher level of dysregulation compared to the control. Similar patterns have been observed in studies of aging and disease, where disrupted network structures have indicated underlying pathology[13,14,28]. We detected no connections between placental histopathological features and any analytes in any condition, preventing our interpretation of their underlying mechanisms or clinical utility. However, we identified a clinically interesting community composed of 100 analytes in control placentas that contained multiple analytes implicated in placental dysregulation, including *FLT1*, *FSTL3*, *HTRA4*, *LEP*, *PHYHIP*, *miR-365a-3p*, and *miR-210-3p*. These genes are substantially dysregulated across the conditions and have disrupted network connectivity within the corresponding FGR + HDP community. Overexpression of these genes is reported in PE placentas[29–33]. Overexpression of *FLT1* and *FSTL3* in preeclamptic placentas have repeatedly been reported using complimentary approaches and distinct patient populations; therefore, they should be given serious consideration as biomarkers for preeclampsia[29,30,32]. In one study, *FLT1* and *FSTL3* alongside additional genes were identified using an unsupervised approach as being highly discriminant between preeclampsia and control placentas[32]. Similarly, in our study we define our networks assuming no prior knowledge. We then narrowed down our focus to a particular subnetwork and further reduced this to the 12 analytes with the highest discriminatory expression. Only at this point did we examine our ability to separate placentas of distinct clinical outcomes in high-dimensional space using an unsupervised approach.

In this community of interest, we found that *miR-365a-3p* was the most connected node in this community in control placentas, indicating that it likely plays a crucial regulatory role within this network that may be essential for healthy placental function. Indeed, *miR-365* upregulation in the decidua has previously been implicated in recurrent miscarriage[34]. HTRA4 plays an important role in trophoblast differentiation into syncytiotrophoblast, and its inability to drive this process in PE placentas could explain the significantly higher extravillous trophoblast and lower syncytiotrophoblast cells observed in FGR + HDP placentas[35]. *miR-210-3p* is hypoxia-induced and has previously been implicated in fetoplacental growth, preeclampsia, and trophoblast dysfunction[31,36–38]. It has also been proposed as a blood serum biomarker for preeclampsia[38]. *miR-210-3p* was the most connected node in the FGR + HDP network, gaining inappropriate network connectivity compared to the control placenta network. Our analysis reveals how these genes interact in a network implicated in hematopoietic progenitor cell differentiation, which is decimated in FGR + HDP placentas. Based on the gene ontology terms, this signal may originate from maternal blood trapped in the placenta at the time of biopsy.

Examining the 100 analytes in this community provided good distinction between Control and FGR + HDP, but less separation was observed between FGR + HDP and FGR, PE, or PTD placentas. This is likely partially explained due to the overlap in clinical characteristics between FGR + HDP and these three other placenta syndromes. By further limiting these analytes to the twelve with the most differential gene expression across all five obstetric conditions, we identified a signature that distinguished FGR + HDP placentas from other conditions despite overlap in clinical features. These twelve analytes were universally upregulated in FGR + HDP placentas compared to all other conditions. They also provided improved separation between FGR + HDP placentas and the other placental syndromes using these twelve analytes in contrast to the full set of 100. This suggests that despite the high level of heterogeneity between the conditions, we were able to identify a signature unique to FGR + HDP placentas.

A limitation of this study is that it uses bulk tissue multiomics and thus cannot distinguish what is happening in specific cell populations other than by estimating with computational inference. We detected a significant difference in the placenta cellular composition in FGR + HDP using deconvolution. Syncytiotrophoblast may be underrepresented with this approach because its multinucleated nature is better captured by single-nuclei sequencing than single-cell sequencing[39]. However, differences in placental cellular composition likely account for some analyte-level differences detected within these conditions. A substantial proportion of *FLT1* and *LEP* overexpression detected in preeclampsia has previously been attributed to placental cellular composition, although their expression levels remain significantly upregulated after adjusting for this[22]. Future research is required to evaluate the biological mechanisms of *LEP*, *FLT1*, and *FSTL3* driving placental dysfunction in FGR + HDP. To elaborate on the mechanism of the *FLT1* / *FSTL3* network, biological validation is required. Finally, placental data is not clinically actionable, all samples in this study are from the end of pregnancy, and the FGR + HDP placental signature is not predictive. We plan in future studies to integrate placenta multiomics data with longitudinal multiomics data collected in other tissues like blood. This will allow us to pair placental functions with predictive measures that can be clinically deployed. It will also improve upon our biological and clinical understanding of the significance of the placental differences between obstetric syndromes reported here. This will account for the complex system of pregnancy, providing a comprehensive view of the patient in their entirety.

Our interomics analyses defined network perturbations across conditions, enhancing our understanding of placental function. Notably, our analyses distinguished clearly between FGR and FGR + HDP at a molecular level, supporting that these may best be considered distinct clinical conditions. This work demonstrates the utility of a systems-biology approach in a single tissue type, setting the stage for more comprehensive pregnancy-related studies across multiple tissue types with longitudinal data. By capturing dynamic molecular transitions throughout pregnancy, such an approach holds promise to uncover key network perturbations that lead to disease. This lays the groundwork for developing predictive, preventive, personalized, and participatory (P4) medicine in obstetric care, ultimately contributing to the broader realization of precision medicine.

## Data availability
The RNA sequencing data that supported the findings of this study have been deposited in the NCBI Sequence Read Archive (SRA) with Bio Project ID PRJNA914646. The metabolomics (DOI: 10.6084/m9.figshare.29640338) and proteomics (DOI: 10.6084/m9.figshare.29640458) data were deposited on Figshare. All processed proteomic, metabolomic, transcriptomic, miRNA, placental histopathology reports, and clinical data used in these analyses are available on GitHub (https://github.com/spiekos/Hood-Lab-Placental-Multiomics/). This includes the data in the raw, cleaned, and adjusted forms. The results of the cellular deconvolution analysis, large supplementary data tables (Supplementary Data 1–7), and condition-specific communities with at least ten nodes are also available on GitHub (https://github.com/spiekos/Hood-Lab-Placental-Multiomics/).

## Code availability
All scripts that performed these analyses are available on GitHub (https://github.com/spiekos/Hood-Lab-Placental-Multiomics/). The versions for all languages and packages used can be found in Supplementary Table 10.

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

## Acknowledgements

We thank Jeannette Wellman, Roya Depasquale, Danielle Sharbaugh, and Tess Capo from the Steve N. Caritis MOMI Database and Biobank for technical assistance. This work was funded by two grants, National Institute for Child Health & Human Development grants (K99HD112600-01, awarded to S.N.P.; R21HD112819, awarded to J.H.) and by grants from the Richard King Mellon Foundation Grant and an anonymous foundation (to Y.S.).

## Author contributions

Conceptualization and methodology, O.B., S.N.P., L.H., N.D.P., and Y.S. Investigation and validation, S.N.P., O.B., T.C., and W.T.P. Formal analysis, S.N.P. and A.B. Visualization, S.N.P. and A.B. Data curation and software, S.N.P. Writing—original draft, S.N.P. Writing—review and editing, S.N.P., O.B., A.B., J.H., L.H., N.D.P., and Y.S. Funding acquisition, S.N.P., O.B., J.H., and Y.S. Resources, project administration, and supervision, L.H. and Y.S. All authors read and approved the final manuscript.

## Competing interests

L.H. and N.D.P. have served as scientific advisors for Sera Prognostics, a pregnancy diagnostics company, and NDP holds stock. The company is not associated with this study or any of the findings. NDP is the Chief Science Officer of Thorne HealthTech and has equity. Y.S. is a consultant at Bio-Rad Laboratories, Inc. All other authors declare no competing interests.

## Ethics

We confirm that all collaborators in this study meet the Nature Portfolio journals' criteria for authorship. Their invaluable contributions to the study's design, implementation, and intellectual output are reflected in their inclusion as authors. Roles and responsibilities were clearly defined and agreed upon by all collaborators prior to the commencement of the research. The research conducted is of significant local relevance, a determination made in close collaboration with our local partners. This study was not severely restricted or prohibited in the primary researchers' setting, and it does not pose any risk of stigmatization, incrimination, discrimination, or personal risk to participants. Local and regional research have been considered and cited appropriately within this manuscript.
