## [Transparent Peer Review file · Communications Biology]

Placental Network Differences Among Obstetric Syndromes Identified With An Integrated Multiomics Approach

Corresponding Author: Dr Samantha Piekos

This manuscript has been previously submitted at another journal. This document only contains information relating to versions considered at Communications Biology.

Version 0:

Reviewer comments:

Reviewer #1

(Remarks to the Author)

The authors provide a multi-omic analysis of placentas from multiple pregnancy complications. The integration and subsequent analyses have allowed for in depth assessment of the network pathways separating pregnancy complications from healthy controls, particularly FGR-HDP, and has identified 12 biomarkers which could be taken forward for further analysis. The conclusions are appropriate to the findings of the study.

Some minor comments about the manuscript:

- In the abstract, there is reference to “detected communities”. For readers who are unfamiliar with network analyses, it may be beneficial to include the phrase “aka subnetworks” which has been included as a description in the results section.
- Placentas were collected from people aged 15-45 years of age. This cohort includes people who are both considered to be young and advanced in maternal age, both of which are more likely to suffer from pregnancy complications. Whilst the correlation matrices in Supplementary Figure 8 shows no correlation between maternal age, the manuscript would benefit from some discussion of this finding as maternal age is perceived to play a role in pregnancy complications in the scientific community.
- Could the authors comment further on their choice of scRNA sequencing sample used for the decomposition of bulk RNA sequencing. Particularly regarding the gestational age of the sample and how this compares to the gestational age of this study’s samples taking into consideration the inclusion of PTD and preeclampsia samples (PTD have a gestational age of <37 week’s gestation, which is commonly also seen in PE). Comment on whether the sample choice would affect the expected cell type proportions in the earlier gestational age samples would also be beneficial.
- Under the section Mining Multiomic Communities, there are presentation of results of “100 analytes” and “12 analytes” and in the legend of Supplementary Figure 20, “100 analytes”, “12 analytes” and “12 mRNA transcripts” are referred to. Would the authors be able to clarify how the 100 and 12 analytes have been selected, and whether the 12 analytes are related to the 12 mRNA transcripts.
- Whilst the definition of PE was taken from the ACOG guidelines, there is currently no information provided on the proportion of patients with PE where the fetus was also FGR. This may in part explain why there is no separation on the PCA in Figure 5 when PE and FGR-HDP are compared. Addition of this information and a comment in the discussion should be added.
- One reference that could be beneficial in discussing this study’s results compared to previously published literature, is that by Leavey et al (2016, <https://www.ahajournals.org/doi/10.1161/hypertensionaha.116.07293>). The study used bulk RNA sequencing and subsequent GSEA analysis and identified four clusters of PE. Discussions of the comparisons between their pre-eclampsia findings, and those of this study may be beneficial, especially as maternal age differences were found between the clusters.
- Whilst figure legends are detailed, it would be beneficial for the number of samples included in the analyses to be stated.
- PLS-DA allowed separation between the samples to be observed, however, the results may be clearer for readers unfamiliar with the analytical approach if the 95% confidence intervals were presented as ellipses in Supplementary Figure 20.
- The use of simulations has produced very convincing results. However, expansion the descriptions of the simulation methodology would be beneficial for less experienced readers.

- Can the authors offer some interpretation of effect sizes in relation to clinical expectations.

Reviewer #2

(Remarks to the Author)

The authors present an important multi-omics study of the placenta highlighting fetal sex and disease-specific changes in the molecular networks. The data availability makes this an important resource in the field. While overall analyses are sound there are a few aspects that need to be improved before publication (see below). Also, ideally, before publication, the github repository should post the multi-omics data in addition to the SRA archive.

Main:

1) Figure 1B seems to show the cell type proportions distributions by disease group (FGR+HDP vs Control). All cell type proportions are marked as significant between the two groups but there is no shift in mean proportions between groups. The authors used a Kolmogorov-Smirnov test which would return always a significant result if the shape of the distributions are different but they have the same mean. I do not believe this is a meaningful analysis. A Wilcoxon test could be used instead and adjustment for multiple cell type testing should be implemented.

2) The authors stated that "There were more significant correlations in the disease networks for PTD, PE, and FGR+HDP by two orders of magnitude compared to the control network after downsampling to $n=30$ to allow evaluation at the same power level. Finding more or less correlations between analytes and gestational age within the different groups depends not only on sample size (as the authors have correctly accounted for), but also by the opportunities to observe changes provided by the gestational age spread. I did not see in the manuscript the information about the gestational age range and mean in each group. While more heterogeneity among patients within disease conditions lead to more significant correlations (as described before e.g. <https://pubmed.ncbi.nlm.nih.gov/26232508/>) the gestational age variability within groups needs to be accounted for.

Minor:

3) "and short and bulk transcriptomics" not clear what short transcriptomics means

4) "and annotated with the latest GENCODE 30" The latest gencode version is 47. Perhaps this could be listed as a minor limitation.

5) The dotted squares in the formulas could be removed to avoid confusion. The squares are the place holder where the variables x should be entered.

6) The use of birthweight percentile <3rd should involve the use severe FGR nomenclature since <10th is the usual definition.

Version 1:

Reviewer comments:

Reviewer #1

(Remarks to the Author)

A comprehensive and clear response. No further questions.

Reviewer #2

(Remarks to the Author)

The authors have addressed my comments and improved their manuscript. I have no further changes to suggest.

We thank the reviewers for their suggestions. We have taken the liberty to individually respond to each reviewer comment individually here in blue. Any changes made to the manuscript in response to reviewer comments are highlighted. We also included reviewer only tables in response to Reviewer 2's request to use a different statistical test to examine cell type proportions between conditions. The main changes to the manuscript are as follows:

1. **Expanded methodological detail:** We added more information to the Methods section, particularly regarding our analytical pipeline, justification for covariate inclusion in regression models, and the deconvolution approach using scRNA-seq data from Campbell et al. (2023). These clarifications enhance the reproducibility and interpretability of our approach.
2. **Improved discussion and contextualization:** We expanded the Discussion to address reviewer concerns, including a deeper exploration of the role of maternal age, the overlap in clinical features between PE and FGR+HDP, and relevant prior studies, such as Leavey et al. (2016).
3. **Clarification of figures and terminology:** We revised several figure legends to include the number of samples, standardized terminology regarding analyte subsets, and updated Supplementary Figure 20 to improve clarity for readers unfamiliar with PLS-DA by incorporating 95% confidence ellipses.
4. **Clarified the rationale for specific analytical decisions:** We added justification for the exclusion of maternal age from our generalized linear models due to multicollinearity with gestational age and explained the implications of using term scRNA-seq data for deconvolving bulk transcriptomic profiles in our samples.

Thanks to the reviewers we have improved the readability and interpretability of our manuscript

Referee expertise:

Referee #1: Placental development

Referee #2: Multiomics data analysis

Reviewers' comments:

Reviewer #1 (Remarks to the Author):

The authors provide a multi-omic analysis of placentas from multiple pregnancy complications. The integration and subsequent analyses have allowed for in depth assessment of the network pathways separating pregnancy complications from healthy controls, particularly FGR-HDP, and has identified 12 biomarkers which could be taken forward for further analysis. The conclusions are appropriate to the findings of the study.

Some minor comments about the manuscript:

- In the abstract, there is reference to “detected communities”. For readers who are unfamiliar with network analyses, it may be beneficial to include the phrase “aka subnetworks” which has been included as a description in the results section.

The reviewer made an excellent suggestion. We added this clarification in the abstract. It now reads: “After adjusting for fetal sex and gestational age, we built condition-specific interomics networks and detected communities (a.k.a. subnetworks).”

- Placentas were collected from people aged 15-45 years of age. This cohort includes people who are both considered to be young and advanced in maternal age, both of which are more likely to suffer from pregnancy complications. Whilst the correlation matrices in Supplementary Figure 8 shows no correlation between maternal age, the manuscript would benefit from some discussion of this finding as maternal age is perceived to play a role in pregnancy complications in the scientific community.

The reviewer is right to highlight the importance of maternal age as an important confounding variable. Our previous publication Barak et al. (2023) summarized the clinical characteristics of the cohorts across the different obstetric syndromes in Table S2, which included maternal age. We therefore did not want to report this information again in this publication but acknowledge that consistent to what has been observed by the scientific community there is a difference in the distribution of maternal ages across the different obstetric syndromes. However, we made sure to update our methods section to better acknowledge the importance of this variable and explain why it was not included as part of our GLMs used to adjust analytes for common confounding variables. We added two sentences to the methods subsection “Adjusting for Fetal Sex and Other Common Covariates” to justify our exclusion of maternal age from our GLMs: “Maternal age is anticorrelated with gestational age (Supplemental Figure 8) and the two variables have a high level of multicollinearity. Therefore, we decided to not include maternal age alongside gestational age in our GLMs.”

- Could the authors comment further on their choice of scRNA sequencing sample used for the decomposition of bulk RNA sequencing. Particularly regarding the gestational age of the sample and how this compares to the gestational age of this study’s samples taking into consideration the inclusion of PTD and preeclampsia samples (PTD have a gestational age of <37 week’s gestation, which is commonly also seen in PE). Comment on whether the sample choice would affect the expected cell type proportions in the earlier gestational age samples would also be beneficial.

Thank you for raising the important point regarding the single-cell RNA sequencing reference used for our deconvolution analysis. We utilized the signature matrix generated by Campbell et al. (2023), which was derived from single-cell data predominantly from healthy term placental villous tissue.

We recognize that our study includes samples from preterm deliveries (PTD) and preeclampsia (PE), which often involve earlier gestational ages than the term samples

used to generate the reference. However, it's important to note that **Campbell et al. (2023) rigorously tested their deconvolution method using bulk RNA sequencing data from preeclampsia samples (GSE75010). This dataset inherently includes samples with earlier gestational ages, reflecting the overlap between PE and PTD.**

Their validation demonstrated robust performance of the deconvolution method in these PE samples, suggesting that the reference is capable of accurately estimating cell type proportions even in samples with gestational age variations. This validation, in addition to their robust *in silico* testing, provides strong support for the applicability of the reference to our study.

While we acknowledge that placental cell type composition can change throughout gestation, and a reference dataset derived from earlier gestational ages might provide even greater accuracy for PTD and PE samples, the validation performed by Campbell et al. (2023) using relevant disease samples mitigates this concern. To further address this point, we added a statement to the manuscript stating more information about how the cell-type gene signature matrix was generated and highlighting the validation of the deconvolution method using both term and preeclampsia samples in the “Deconvolving Bulk RNA-seq Data” subsection of the Methods.

- Under the section Mining Multiomic Communities, there are presentation of results of “100 analytes” and “12 analytes” and in the legend of Supplementary Figure 20, “100 analytes”, “12 analytes” and “12 mRNA transcripts” are referred to. Would the authors be able to clarify how the 100 and 12 analytes have been selected, and whether the 12 analytes are related to the 12 mRNA transcripts.

Additional information was added to the Methods subsection of “Mining Multiomic Communities For Biological Signatures” as well as the Discussion section. The figure legend fro Supplemental Figure 20 was also updated to match the terminology in the main manuscript. These changes were made to address both this and the following comment made by the reviewer. In short indeed “12 analytes” in the main paper and the “12 mRNA transcripts” referred to in the original Supplemental Figure 20 legend are one and the same.

We selected a community that we identified in control placentas based on the presence of several analytes previously implicated in the literature of placental dysfunction. This community contained 100 analytes, which we then focused on for the next section of the paper in which we compared the connectivity and the differential analyte levels between the various obstetric syndromes. The twelve analytes with the most differential gene expression across all pairwise comparisons between obstetric syndromes were then identified as a signature using hierarchical clustering. These twelve analytes were all upregulated in FGR+HDP placentas compared to every other syndromes. We then compared FGR+HDP placentas to all other obstetric syndromes in high dimensionality space using both the 100 analytes that composed the entire control community and

separately using just the 12 analytes (all mRNA transcripts) that were identified as having the strongest signal of differential levels between the obstetric syndromes

- Whilst the definition of PE was taken from the ACOG guidelines, there is currently no information provided on the proportion of patients with PE where the fetus was also FGR. This may in part explain why there is no separation on the PCA in Figure 5 when PE and FGR-HDP are compared. Addition of this information and a comment in the discussion should be added.

The reviewer makes an astute observation. Of the patients that were classified with FGR+HDP all of them were in <3rd percentile for gestational age and had pregnancy related hypertension. Of the 30 patients that were in this group 26/30 (86.7%) also have preeclampsia. We have added this information to the “Study participants” subsection of the Methods. We agree with the reviewer that this overlap in clinical outcomes between the FGR+HDP and PE groups could partially explain why there was poorer separation between the two groups in the PCAs included in Figure 5. We have added more detailed discussion diving into the heterogeneity and our attempt here to disambiguate the biological signal between the obstetric syndromes.

Heterogeneity within clinical maternal and perinatal outcomes remains a challenging problem in our understanding of these diseases and their overlapping clinical features and outcomes. By examining multiple outcomes simultaneously here we attempt to disambiguate the high degree of heterogeneity in clinical presentation. We show that through limiting the multi-omic community (subnetwork) of interest to the analytes with the most distinctive differential levels between the obstetric syndromes that we are able to better distinguish between these heterogeneous syndromes – despite an overlap in clinical outcomes - than using the entire network. This suggests that we are indeed finding distinctive biological signal unique to FGR+HDP placentas and that this can serve as a signature unique to this syndrome.

- One reference that could be beneficial in discussing this study’s results compared to previously published literature, is that by Leavey et al (2016, <https://www.ahajournals.org/doi/10.1161/hypertensionaha.116.07293>). The study used bulk RNA sequencing and subsequent GSEA analysis and identified four clusters of PE. Discussions of the comparisons between their pre-eclampsia findings, and those of this study may be beneficial, especially as maternal age differences were found between the clusters.

We thank the reviewer for sharing this relevant paper with us. We made sure to discuss this paper in the context of our papers approach. We made sure to highlight the distinct experimental differences in the approaches. This adds value that both studies identified FLT1 and FSTL3 as being of particular interest to distinguishing preeclamptic placentas as the studies use different methodological approaches and apply PCA at different points. Our study in particular does not assume any prior knowledge, which is valuable because gene sets were not developed in the context placenta biology or obstetric syndromes, which is a drawback in using them to interpret placenta data.

- Whilst figure legends are detailed, it would be beneficial for the number of samples included in the analyses to be stated.

The reviewer makes a great suggestion to make sure to consistently include sample size in all of the figure legends. We have gone ahead and done this. Changes to the figure legends are highlighted.

- PLS-DA allowed separation between the samples to be observed, however, the results may be clearer for readers unfamiliar with the analytical approach if the 95% confidence intervals were presented as ellipses in Supplementary Figure 20.

The reviewer makes a great suggestion. We have updated Supplementary Figure 20 to include the 95% CI – indicated by the dotted line ellipses around each group. This should help the reader interpret the plots better. We also updated the corresponding code for these plots on our GitHub so our code reflects this change.

- The use of simulations has produced very convincing results. However, expansion the descriptions of the simulation methodology would be beneficial for less experienced readers.

We appreciate the reviewer's suggestion to make the methodology of this paper more accessible to the reader. We therefore added a couple of sentences explaining at a high level how random simulation functions as a statistical test and its purpose in this paper. These sentences were added to the "Adjusting for Fetal Sex and Other Common Covariates" subsection of the Methods section as follows: "This simulation approach effectively creates a null distribution, representing what the overlap of gestational age-regulated analytes would look like if there were no true sex-specific differences. By comparing our actual results to this null distribution, we can assess the statistical significance of our findings and estimate the FDR."

- Can the authors offer some interpretation of effect sizes in relation to clinical expectations.

We thank the reviewer for querying about some of the effect sizes in our data in the context of clinical applications. While many of the statistically significant effect sizes that we reported were relatively small, some were marked (e.g., EVT_s between FGR+HDP vs control in 1B, and Suppl Fig 4, and the interomics correlations in Suppl. Fig 12). It might not be surprising that some of the effect sizes are not striking in the clinical context, as there is a significant overlap in the pathobiology of these syndromes. Hence, as our goal was to harness the power of interomics to improve our understanding of clinical syndromes, not to build networks that might serve for diagnostics, we would hesitate to make clinical inferences. We project that further refining our interomics approach using prospectively collected data across pregnancy might advance our omics-based definition of gestational syndromes and possibly propose actionable approaches to diagnostics, therapeutics or disease prevention. We have added the

following sentence to the limitations paragraph – second to last paragraph in the Discussion – to better relay this intent: “It will also improve upon our biological and clinical understanding of the significance in the placental differences between obstetric syndromes reported here.”

Reviewer #2 (Remarks to the Author):

The authors present an important multi-omics study of the placenta highlighting fetal sex and disease-specific changes in the molecular networks. The data availability makes this an important resource in the field. While overall analyses are sound there are a few aspects that need to be improved before publication (see below). Also, ideally, before publication, the github repository should post the multi-omics data in addition to the SRA archive.

Main:

1) Figure 1B seems to show the cell type proportions distributions by disease group (FGR+HDP vs Control). All cell type proportions are marked as significant between the two groups but there is no shift in mean proportions between groups. The authors used a Kolmogorov-Smirnov test which would return always a significant result if the shape of the distributions are different but they have the same mean. I do not believe this is a meaningful analysis. A Wilcoxon test could be used instead and adjustment for multiple cell type testing should be implemented.

We thank the reviewer for the suggestion of an alternative statistical test. While the reviewer is correct that the Kolmogorov-Smirnov test is sensitive to the shape as well as the location of the distribution, the proposed alternative test (Mann-Whitney U test aka Wilcoxon rank-sum test) has the same alternative and null hypotheses as the Kolmogorov-Smirnov test and is therefore, by definition, testing the same thing and plagued with the same subtleties. One can see this by looking at both the original 1933 paper by Kolmogorov (pg 139 in the attached book) and the original 1947 paper by Mann and Whitney (attached) where the tests are proposed. Neither states an alternative hypothesis of different means or medians. Only when one adds the assumption that the distributions are the same shape can one claim that there is a difference in medians between the populations, although one can still not claim anything regarding the mean. That being said, these tests do test for stochastic dominance — that is whether the CDF of one population is always larger than the CDF of the other. For our purposes this should be sufficient, since even if the means or medians are equal, if the probability of observing a particular value x in one population is always larger than observing the same value in the different population for every value x , then this, by any reasonable measure, is of biological and medical relevance, albeit in a more subtle manner than the standard “the mean in one sample is less than the mean in another”.

Still, we performed the requested analysis and found that there are more significant differences in distributions using the Mann-Whitney test than the Kolmogorov-Smirnov test. Thus, the Mann-Whitney U test is, contrary to the reviewer’s concern, a more

liberal test than the KS test, possibly owed to the use of rank statistics instead of the empirical CDF itself. We have included the requested analysis as a reviewer only tables so that the reviewer can evaluate the results themselves.

2) The authors stated that “There were more significant correlations in the disease networks for PTD, PE, and FGR+HDP by two orders of magnitude compared to the control network after downsampling to n=30 to allow evaluation at the same power level. Finding more or less correlations between analytes and gestational age within the different groups depends not only on samples size (as the authors have correctly accounted for), but also by the opportunities to observe changes provided by the gestational age spread. I did not see in the manuscript the information about the gestational age range and mean in each group. While more heterogeneity among patients within disease conditions lead to more significant correlations (as described before e.g. <https://pubmed.ncbi.nlm.nih.gov/26232508/>) the gestational age variability within groups needs to be accounted for.

We agree with the reviewer that gestational age is an important confounding variable that must be taken into account to interpret the results of this paper. Which is why we had already adjusted the analyte values for gestational age as a function of fetal sex prior to conducting our network analyses. This adjustment was described in our “Adjusting for Fetal Sex and Other Common Covariates” subsection of our Methods. To make the fact that we performed this adjustment for gestational age we added clarifying sentences in both our Methods (“Correlation Network and Community Analysis” subsection) and Results section (“community Structure In Interomics Correlation Networks” subsection).

Minor:

3) “and short and bulk transcriptomics” not clear what short transcriptomics means.

Thank you for pointing out the ambiguity in our terminology. We apologize for the lack of clarity. By 'short transcriptomics,' we were referring to **microRNA sequencing (miRNA-seq)**. We revised the manuscript to clarify that 'short transcriptomics' is microRNA-sequencing

4) “and annotated with the latest GENCODE 30” The latest gencode version is 47. Perhaps this could be listed as a minor limitation.

The reason why we used GENCODE 30 is because this is the version of GENCODE used by our mRNA alignment software STAR when we did the analysis. We have removed “the latest” from our description in the Methods.

5) The dotted squares in the formulas could be removed to avoid confusion. The squares are the place holder where the variables x should be entered.

We thank the reviewer for pointing out this formatting error to us! We have updated the manuscript to ensure that the formula properly shows without any formatting issues.

6) The use of birthweight percentile <3rd should involve the use severe FGR nomenclature since <10th is the usual definition.

The reviewer is correct to point out that our requirement of birthweight percentile <3rd means that we are selecting the placentas with FGR meets the clinical definition of severe fetal growth restriction (FGR). We have added this qualifier when introducing the obstetric syndromes FGR and FGR+HDP in both the abstract and the introduction of our paper.

The reviewers requested no further changes after this round of reviews. We focused on meeting Nature publication formatting guidelines at the request of the editor. We have responded to all editorial requests.

REVIEWERS' COMMENTS:

Reviewer #1 (Remarks to the Author):

A comprehensive and clear response. No further questions.

Reviewer #2 (Remarks to the Author):

The authors have addressed my comments and improved their manuscript. I have no further changes to suggest.